# Creativity Assessment in Subjects with Tourette Syndrome vs. Patients with Parkinson’s Disease: A Preliminary Study

**DOI:** 10.3390/brainsci7070080

**Published:** 2017-07-09

**Authors:** Carlotta Zanaboni Dina, Mauro Porta, Christian Saleh, Domenico Servello

**Affiliations:** Tourette Syndrome Centre, Clinical and Research Galeazzi Hospital, Milan 20161, Italy; mauroportamilano@gmail.com (M.P.); chs12us75010@yahoo.com (C.S.); servello@libero.it (D.S.)

**Keywords:** creativity, tic, Tourette syndrome, Parkinson disease

## Abstract

(1) Background: Literature suggests that high levels of dopamine are associated with creative thoughts. Tourette Syndrome (TS) patients have high dopamine levels, while Parkinson’s Disease (PD) subjects have low dopamine levels. Consequently, TS individuals are supposed to have a major and PD patients less creative output. Moreover, dopamine medications may alter the level of creativity, and therefore Quality of Life, in both pathologies. (2) Methods: The aim of the study was to verify the hypothesis of TS patients having higher creative scores than PD patients. The assessment consisted of the administration of the Creative Thinking ASK Test. There were 54 participants—36 males and 18 females—i.e., 27 TS patients and 27 PD subjects. Age of the sample was 35 to 57 years old, high school certificate was required. (3) Results: TS sample (103.11 ASK average score) was more creative than PD sample (94.11 ASK average score). (4) Conclusions: The results supported the aforementioned hypothesis: TS sample resulted in having higher creative scores than PD sample. Dopamine and other neurotransmitters of TS and PS appear to affect subject’s creativity. Further studies with creative assessments in TS and PD patients are needed to support the preliminary results of our study.

## 1. Introduction

### 1.1. Tourette Syndrome

Once believed to be a rare disease, Tic Disorders [1]—including Tourette Syndrome—make up 1% of the population, and up to 20% in childhood [2].

The disorder mainly affects males by a male to female ratio of 3–4:1; TS begins before age 18, usually around age 6 [3]. In youth, the disorder often increases in severity peaking in the prepubertal period, while in adulthood, in 50–75% of cases, tics improve dramatically [4,5,6,7].

Recently, TS studies showed high correlations of tics with behavioural disorders such as Attention Deficit Hyperactivity Disorder (ADHD), Obsessive-Compulsive Behaviours/Disorders/Symptoms (OCB/ OCD/OCS), anxiety, depression and poor impulse control e.g., SIB (Self-Injurious Behaviour) and NOSI (Non-Obscene Socially Inappropriate behaviour) [8].

Anatomically, basal ganglia and substantia nigra are considered the neuroanatomical regions associated with Tourette Syndrome (TS), Parkinson’s Disease (PD) and other movement disorders. Whereas, from an etio-pathological point of view an alteration of dopaminergic synaptogenesis appears to determine TS symptoms [9].

Many famous artists were affected by TS, such as Samuel Johnson [10]. Several authors [11,12,13,14] examined the possible TS diagnosis of Mozart, hypothesising that the syndrome actually contributed to the musician’s creativity. However, it remains unclear if Mozart was truly affected by TS. His behaviour could have been due to a wide spectrum of neurobehavioural disorders. Zanaboni et al. [15] analysed the creative personality of a sample of 23 Italian TS subjects between 6 and 18 years old and of a healthy control group through Williams’ Creativity test; by results, TS subjects obtained greater creative scores compared with controls in the Flexibility subtest. Wei [16] studied creativity in a group of 127 Taiwanese TS children and a control group of 138 Taiwanese by parents’ reports. In the Elaboration subtest of Creative Thinking, TS patients were less creative than controls, whereas most TS parents reported their children to have high academic achievement.

Creativity in TS adults has never been studied in literature; for this reason, our group decided to pursue this issue in adulthood.

TS dopaminergic theory is supported by the observation of reduction or suppression of symptoms—and creativity skill, as said above—in a substantial number of patients using dopamine receptor blockers such as haloperidol or pimozide. Also, tetrabenazine, a dopamine-depleting drug that inhibits the vesicular monoamine transporter 2 (VMAT2), is effective for various hyperkinetic movement disorders, including TS [17,18]. Results from a study on TS subjects show that after two years of tetrabenazine therapy, 80% of patients have a long-term improvement of symptoms [19]. However, more recent data are suggesting a still more complex picture, whereby also alterations in the GABAergic system may play an important role in the pathophysiology of TS [17].

### 1.2. Parkinson Disease

Parkinson Disease is one of the leading causes of neurological disability for individuals over 60 years of age. PD impacts an estimated 4.5 million people across the world’s 10 most populous countries [20].

In PD there is a loss of dopaminergic neurons in the pars compacta of the substantia nigra (SNpc) with the presence of “Lewy bodies” due to misfolded aggregated alpha-synuclein protein.

The “cardinal signs” of PD are resting tremor, rigidity, bradykinesia-akinesia, postural instability and one-sidedness of the symptoms [21]. Patients can show freezing, i.e., the inability to start or carry on a movement [22]. Moreover, during complex and automated tasks, i.e., writing or a speech, motor control mechanisms can be lacking [23].

Also in TS subjects, complex and automatic movements can be lacking and can, therefore, generate a variety of tics, such as writing or walking or speaking or reading tics.

PD non-motor symptoms include urinary diseases issues, constipation, skin problems, sleep difficulties, cramps and paresthesia, cognitive decline, depression and other emotional changes. Depression, anxiety and emotional changes may also occur in TS patients. Actually, in a longitudinal study by Lin et al. [24] stress levels and tic severity were evaluated in a sample of 45 children with TS and/or OCD, and in a sample of 41 healthy control children. In general, psychosocial stress level was higher in patients with TS compared with the control group. Moreover, those who had higher levels of stress were more likely to have a more severe disease course and major complications. The Social Impairment component is therefore essential for TS assessment, considering that exacerbation of tics is often preceded by stressful life events.

PD treatment is principally pharmacological, i.e., dopaminergic therapy. Moreover, PD surgical treatment, i.e., Deep Brain Stimulation (DBS), is reserved for severe and refractory patients [25,26]. 

More recently, Automated Mechanical Peripheral Stimulation (AMPS) is used in PD patients to treat motor deficits. This orthopedic device repetitively and automatically stimulates specific points of the foot to activate brain areas involved in the management of motor functions [27].

### 1.3. Creativity

There is a plethora of definitions for creativity [28]. According to Tudor Powell Jones [29], creativity is the combination of flexibility, originality and ability to adopt readily new and unusual cognitive patterns in order to solve problems.

Williams’ creativity model, the one used at Milan Tourette Syndrome Centre, defines creativity as the ability to generate ideas or innovative artistic products.

For several years, researchers such as Flaherty [30] have hypothesised that high dopamine levels in the mesolimbic area improves working memory, facilitating mental connections and original thoughts.

Among the different studies [31,32], Limb and Braun [33] of the Johns Hopkins University School of Medicine in Baltimore decided to monitor the brain activity of six jazz pianists in order to find out the brain areas of creative improvisation. The authors concluded that the ability to improvise (the “peak” of creativity and self-expression) appears to be the result of a reduction of self-control and planning in the brain.

Moreover, in our clinical experience, TS patients have a strong predisposition for the arts (music, painting, expressive writing and theatre) and for jobs requiring creativity and divergent thinking.

The literature reports many cases of people at the onset of their PD who developed increased artistic skills apparently secondary to dopamine drugs [34]. Furthermore, according to the neurologist Israeli R. Inzelberg [35], patients with PD assuming dopamine suddenly manifested creative activities. In addition, an Italian research group [36] hypothesised that dopaminergic therapy could encourage creative thinking in some patients with PD.

Differently, Canesi et al. [37] evaluated whether artistic production and creative thinking are influenced by dopamine therapy or artistic skills in a sample of (1) professional artist PD patients (2) PD patients with artistic production (3) PD patients without artistic production. These groups were compared with healthy controls. Creativity scores were higher in healthy professional artists and in professional artist PD patients. PD patients without artistic production and healthy control without artistic production showed creative scores without significant difference. Creativity scores were not significantly correlated with dopamine dosage and duration. Authors suggested that further studies in “on” and “off” medication of PD patients are needed to understand these controversial results.

In this sense, fundamental appears to be the type and dose of medication of the patient because, as mentioned above, the intake of dopamine agonists can increase not only the frequency of being engaged in artistic activities, but also it changes the artistic quality of patients’ creative products.

To conclude, creativity manifests itself differently in TS and PD. These diseases affect people of different ages: TS mostly affects children and adolescents [3], while PD affects adults and the elderly [20]. Based on the available data, we suppose that individuals with TS have a creative personality predisposition, unlike PD subjects who may be compromised in their creativity because of reduced production of dopamine caused by the disease. Therefore, the aim of the study is to verify the hypothesis of TS group of patients having higher creative scores than PD group of patients.

## 2. Materials and Methods

### 2.1. Sample

Subjects are patients of the Tourette Syndrome Centre of Milan Clinical and Research Galeazzi Hospital and of the Parkinson’s Disease and Movement Disorders Unit of Pavia Mondino National Neurological Institute; patients were assessed between January 2014 and July 2016.

All participants signed the informed consent to undergo the test administration and to communicate their sensitive data for research purposes. Project identification number is n.1 of 17th November 2014. Ethics Committee was not involved as the study consisted only in a written assessment test; treatments did not vary due to the study.

The sample follows these socio-demographic characteristics:Age between 35 and 57 years old (see Table 1, Table 2 and Table 3);Italian subjects of both genders (see Table 4);Minimum schooling requirements: high school certificate.

Inclusion criteria are:Subjects with the above-listed socio-demographic characteristics;Subjects who accepted to undergo the test administration;Diagnosis given by a specialist neurologist with diagnostic scales, i.e., YGTSS [38] in TS patients and UPDRS [39] in PD patients (see Table 1, Table 2 and Table 5);Patients in any certified treatment, whether no treatment, psychotherapy, drugs or DBS (see Table 1 and Table 2). Drug types are heterogeneous in the sample. Patients in psychotherapy were undergoing Cognitive Behavioural Psychotherapy [8]. DBS targets in TS patients were Anterior Limb of the Internal Capsule/bed nucleus stria terminalis or Ventral oral anterior-CentroMedian/Parafascicular complex. N.B. Deep Brain Stimulation in TS patients was used to treat Obsessive-Compulsive Behaviours. DBS targets in PD patients were SubThalamic Nucleus or posterior Globus Pallidus interna.

Exclusion criteria are:The patient has not concluded the test, even partiallyOther psychiatric disorders (not considering TS and PD comorbid psychopathologies)

### 2.2. Age and Gender of the Sample

Sample average age is (Table 3) 47.48 years old (SD = 8.16); in particular, in PD subjects (Table 3) the average age is 53.03 (SD = 5.09), whereas in TS subjects (Table 3) the average age is 41.92 (SD = 6.77).

Subjects are mostly males (*n* 36 males versus *n* 18 females, see Table 4). Actually, Tourette Syndrome is mostly diffused between males [3].

TS patients’ average YGTSS score is (Table 5) 46.14 (SD = 14.07); whereas PD patients’ average UPDRS score is (Table 5) 89.81 (SD = 26.38).

### 2.3. Test ASK

ASK Test by Schuler and Hell [40] Italian adaptation by Faraci and Clarotti [41] has been used to evaluate the level of creativity of the sample. The test has been chosen as it is the only available Italian test assessing creativity factor in adults. 

This tool is designed to analyse how individuals face and solve problems. It includes two sections, which can be used together or separately with no limitations if separately [41,42]:“Inferential Thinking” subtest, which investigates the analytical and logical thinking;“Creative Thinking” subtest, which investigates the creative personality.

For the present research, the participants were administrated the single subtest “Creative Thinking” because the other subtest includes graphic items and it is therefore difficult to be completed by patients with writing tics cf. TS patients [43] or with agraphia, impaired handwriting and micrographia cf. PD patients [44].

Assuming the 10 points of D.S. reported in Literature [41], we clinically hypothesised a difference of 9 points between PD and TS groups with an alpha error at 0.05 (5%) and a 0.10 beta error (90% power of the test).

Considering these parameters, the necessary sample size was found to be equal to 54 subjects in total: 27 PD subjects and 27 TS subjects.

ASK test was administered individually and patients were assisted in case of difficulties in writing (PD patients) or writing tics (TS patients). Another difficulty in the execution of the test is related to the ability to concentrate for some PD and TS patients, no measures were taken for this issue.

Subtest “Creative Thinking” consists of the following tasks:“Inventing sentences”“Producing hypotheses”“Defining the conditional structure”“Creating categories”

### 2.4. Use of the Test

The administration took about 40 minutes for each subject. For each patient, ongoing treatment has been documented (Table 1 and Table 2).

Each valid response obtained 1 point as score.

For each task, a total raw score has been calculated. Therefore, each patient had 5 scores: 4 task scores corresponding to the results of “Inventing phrases”, “Producing hypotheses”, “Defining the conditional structure”, “Creating categories”, and a total score for the entire ASK test “Creative Thinking”.

Statistical nonparametric test employed is Mann–Whitney U test. The aim was to verify whether TS group score is statistically higher than PD group score in the tasks and in the total score, and if there is any other statistical difference considering gender and age.

## 3. Results and Discussion

### 3.1. Average ASK Results of the Sample

Table 6 shows average scores of TS group and PD group in the four tasks and in the total score, and Mann–Whitney U Test analysis of the scores. Patients with TS achieved an average total score of 103.1 ± 13.0, while patients with PD obtained an average total score of 94.4 ± 10.0: the difference between the two total scores is statistically significant (*p* = 0.019, see Table 6), and clinically significant as TS group score is 8.7 points higher than PD group score (see Par. Method, Subpar. Test ASK).

Also, the difference between task 1 “Inventing Sentences” (PD = 93.6 ± 7.4; TS = 99.7 ± 9.7) and task 2 “Producing Hypotheses” (PD = 95.5 ± 8.2; TS = 102.1 ± 8.3) average scores of PD and TS patients is statistically significant (*p* = 0.011 and *p* = 0.003, see Table 6).

Contrarily, the difference of the two groups in Task 3 “Defining the conditional structure” and in Task 4 “Creating categories” is not statistically significant.

Given these results, it is possible to give credit to the study hypothesis: patients with TS exhibit higher creative scores when compared with individuals with PD.

### 3.2. ASK Results of the Sample in Score Ranges

ASK manual reports three score ranges for the subtest “Creative Thinking”: low score range (below 93 points); average score range (between 93 and 07 points); high score range (over 107 points).

The study analysis has been carried out on 54 subjects, i.e., 27 patients with PD and 27 patients suffering from TS. The number of subjects with PD who have obtained a score of less than 93 points is *n* = 14 (51.9% of the subjects with PD) compared with *n* = 5 subjects in TS group, that is, 18.5% of subjects with TS (Table 7).

The number of PD subjects achieving a score between 93 and 107 points is *n* = 10 (37% of the subjects with PD), compared with *n* = 14 subjects in TS group, that is, 51.9% of patients with TS (Table 7).

Finally, the number of subjects with PD who scored higher than 107 points is *n* = 3 (11.1% of the subjects with PD) compared with *n* = 8 in TS group, that is, 29.6% of patients with TS (Table 7).

Therefore, individuals with PD and TS are differently positioned in the three ranges of scores: in particular, those with TS are positioned on average in the upper range, i.e., resulting in over 107 points, while subjects with PD are placed on average in the lower range, i.e., resulting below 93 points (Table 7).

At last, Table 6 and Table 7 give credit to the initial hypothesis: TS patients result in having higher creative scores than PD patients.

Considering other possible results, no significant differences concerning creative assessments in the two groups have been found in relation to age and sex.

### 3.3. Discussion

Although the study hypothesis has been confirmed, many considerations are needed about strengths and limitations of the study itself.

First, mean age difference between patients is one of the limitations of the study. The limitation is due to the different age of development of the two diseases: childhood and adolescence of TS and adulthood of PD. We opted for a range age of 35–57 years old in order to terminate the study in approximately two years (2014–2016).

Secondly, considering the ongoing treatments, the sample was too heterogeneous (type of drugs, DBS targets, DBS settings) to produce a comparison between subjects. Researchers have therefore been limited to note the type of intervention in progress. Therapy difference between subjects is a limitation of the study because they could have had an effect on the sample’s dopamine levels and, therefore, on the creativity assessment. However, it remains unclear in which way treatments had modulated creativity in each pathology. Probably, medications worked against the study hypothesis because antidopaminergic treatments in TS group should have reduced creativity while reducing symptoms.

However, as PD and TS patients usually undergo heterogeneous therapies, our sample is considered a “reachable sample” in an ordinary hospital. Again, we opted for a sample with heterogeneous therapies because homogeneous therapy groups of patients would have meant a longer period of recruitment and/or the necessity of involving multiple centres in the study.

Geographical area of recruitment was limited to Italy, and patients were coming from only two centres. We decided to concentrate this study at the Galeazzi Hospital and the Mondino Institute.

For the above-mentioned limitations, the study is considered as a preliminary study.

## 4. Conclusions

This study supported our hypothesis of highlighting a predisposition to creativity in patients with TS rather than in PD patients. Actually, TS creative scores were higher than PD creative scores; therefore, the aim of the study has been reached.

During the clinical assessment, creativity can be a useful element for the differential diagnosis of TS and PD. Consequently, further studies are needed about creativity in TS even in childhood and adolescence.

Considering interventions, the psychosocial intervention for both patients with TS and PD should take into account the creative factor to be further stimulated in TS, and to be increased in PD. Art therapy, including music therapy and theatre, are just a few examples of psychosocial intervention to be considered in TS and PD patients.

The main limitation of the study consisted in the comparison of a neurodevelopment disorder (TS) with a neurodegenerative disorder (PD) because of the significant age difference of the two groups, and therefore the lack of an age-matched sample. This limitation leads to preliminary results that need to be integrated with results of bigger samples and/or with a smaller age range of the sample. Furthermore, the heterogeneity of therapies of patients is another limitation for the comparison of the two groups because of the possible effect of therapies on patients’ creative assessment. Actually, it is possible that the medication intake works against our hypothesis: TS group receives medications that block dopaminergic activity and, as a result, creativity should also diminish. Again, this limitation leads to preliminary results that need to be integrated with results of homogeneous therapy groups of patients. For these reasons, the current study can be considered a preliminary study. Conversely, the absence of any literature comparing creative traits in PD and TS is the principal strength of the current study.

Further multi-centre studies are needed with larger samples of age-matched subjects and/or from different geographic areas and/or with single or multiple homogeneous therapies, in order to extend our preliminary results. A study on the discussion of different comorbidities of TS in relation to creativity will be realised by our group in the future.

## Figures and Tables

**Table 1 brainsci-07-00080-t001:** Age, The Yale Global Tic Severity Scale (YGTSS) [38] score and ongoing treatments of Tourette Syndrome (TS) sample. * DBS = Deep Brain Stimulation.

TS Patient N	Age (years) at the Test Administration	YGTSS [38] Score at the Test Administration	Ongoing Treatments (No Treatment-Psychotherapy-Drugs-DBS *)	Type of Drugs and Posology
1	42	45	Drugs + DBS	aripiprazole 15 mg − quetiapine 25 mg
2	56	40	No treatment	
3	44	68	Psychotherapy + drugs	alprazolam 0.5 mg
4	57	26	No treatment	
5	44	20	No treatment	
6	43	31	No treatment	
7	37	50	Drugs + DBS	aripiprazole 15 mg
8	35	27	No treatment	
9	51	63	Drugs	escitalopram 10 mg − alprazolam 0.5 mg
10	41	49	Drugs	aripiprazole 15 mg − topiramate 25 mg
11	35	51	Drugs	aripiprazole 15 mg − fluvoxamine 100 mg
12	35	28	No treatment	
13	46	62	Drugs	clomipramine 75 mg − ketazolam 30 mg × 2 die
14	41	33	No treatment	
15	38	41	Drugs	tetrabenazine 25 mg × 3 die
16	37	48	Drugs + DBS	aripiprazole 15 mg
17	35	57	Drugs	risperidone 1 mg × 2 die
18	35	59	Drugs	pimozide 4 mg − aripiprazole 25 mg
19	50	61	Drugs	citalopram 25 mg
20	49	64	Drugs	topiramate 25 mg − tiapride hydrochloride 100 mg × 2 die − duloxetine 30 mg
21	46	48	Drugs	aripiprazole 15 mg − clomipramine 25 mg
22	37	51	Drugs	pimozide 4 mg
23	44	60	Drugs	pimozide 4 mg
24	35	23	No treatment	
25	35	52	Drugs	pimozide 4 mg
26	49	34	No treatment	
27	35	55	Drugs	pimozide 4 mg

**Table 2 brainsci-07-00080-t002:** Age, Unified Parkinson’s Disease Rating Scale (UPDRS) [39] score and ongoing treatments of PD sample.

PD Patient N	Age (Years) at the Test Administration	UPDRS [39] Score at the Test Administration	Ongoing Treatments (Drugs-DBS *)	Type of Drugs and Posology
1	56	96	Drugs	pramipexole ER 0.26 mg
2	57	68	Drugs + DBS	ropinirole 4 mg − rasagiline 1 mg
3	40	58	Drugs + DBS	ropinirole 4 mg − rasagiline 1 mg
4	57	102	Drugs	rasagiline 0.5 mg − pramipexole ER 0.26 mg − carbidopa hydrate 27 mg − melevodopa hydrochloride 314 mg
5	54	99	Drugs	levodopa 100 mg − carbidopa 25 mg − fluvoxamine 100 mg
6	57	75	Drugs + DBS	levodopa 50 mg × 4 die − benserazide 12.5 mg × 4 die
7	57	122	Drugs	levodopa 100 mg × 4 die − carbidopa 25 mg × 4 die − entacapone 200 mg × 4 die
8	48	118	Drugs	levodopa 100 mg × 3 die − carbidopa 25 mg × 3 die − entacapone 200 mg × 3 die − rasagiline 1 mg − thiamine 300 mg × 2 die − folic acid 5 mg
9	52	61	Drugs + DBS	carbidopa hydrate 27 mg − melevodopa hydrochloride 125.6 mg − rotigotine 2 mg + duloxetine 60 mg + folic acid 5mg
10	46	64	Drugs + DBS	levodopa 100 mg × 3 die − benserazide 25 × 3 die − rotigotine 4 mg
11	56	139	Drugs	levodopa 75 mg × 4 die − carbidopa 18.75 mg × 4 die − entacapone 200 mg × 4 die − fluvoxamine 50 mg − rotigotine 8 mg
12	57	116	Drugs	carbidopa hydrate 13.5 mg − melevodopa hydrochloride 157 mg − pramipexole ER 0.26 mg − levodopa 100 mg − carbidopa 25 mg − rasagiline 1 mg
13	56	131	Drugs	levodopa 75 mg × 3 die − carbidopa 18.75 mg × 3 die − entacapone 200 mg × 3 die − carbidopa hydrate 27 mg × 4 die − melevodopa hydrochloride 125.6 mg × 4 die, ramipril 2.5 mg, venlafaxine 75 mg
14	44	52	Drugs + DBS	ropinirole 4 mg − rasagiline 1 mg
15	57	97	Drugs	levodopa 200 mg × 3 die − benserazide 50 mg × 3 die − rasagiline 1 mg − ropinirole ER 8 mg − mirabegron 50 mg
16	55	111	Drugs	pramipexole 0.7 mg − levodopa 125 mg × 3 die − carbidopa 31.25 mg × 3 die − entacapone 200 mg × 3 die
17	56	73	Drugs	pramipexole 0.52 mg − pramipexole 0.26 × 2 die
18	46	81	Drugs	rotigotine 4 mg − levodopa 100 mg × 4 die − carbidopa 25 mg × 4 die − entacapone 200 mg × 4 die
19	56	101	Drugs	levodopa 100 mg × 4 die − carbidopa 25 mg × 4 die − entacapone 200 mg × 4 die
20	57	49	Drugs + DBS	carvedilol 25 mg
21	56	114	Drugs	levodopa 75 mg × 3 die − carbidopa 18.75 mg × 3 die − entacapone 200 mg × 3 die − carbidopa hydrate 27 mg × 4 die − melevodopa hydrochloride 125.6 mg × 4 die
22	57	58	Drugs + DBS	ropinirole 4 mg − rasagiline 1 mg
23	44	67	Drugs	rasagiline 1 mg
24	51	75	Drugs	pramipexole 0.52 mg − duloxetine 30 mg
25	56	111	Drugs	levodopa 100 mg × 3 die − benserazide 25 mg × 3 die − rasagiline 1 mg
26	49	70	Drugs	pramipexole 1.05 mg
27	55	117	Drugs	levodopa 50 mg × 4 die − benserazide 12.5 mg × 4 die

* DBS = Deep Brain Stimulation.

**Table 3 brainsci-07-00080-t003:** Sample average age and standard deviation.

	N° of Subjects (%)	Average Age	Standard Deviation
Parkinson Disease sample	27 (50%)	53.03	5.09
Tourette Syndrome sample	27 (50%)	41.92	6.77
Total sample	54 (100%)	47.48	8.16

**Table 4 brainsci-07-00080-t004:** Gender of the sample.

	Diagnosis		Total
Gender	Parkinson Disease	Tourette Syndrome	
Male	20 (55.6%)	16 (44.4%)	36 (100%)
Female	7 (38.9%)	11 (61.1%)	18 (100%)
Total sample	27 (50%)	27 (50%)	54 (100%)

**Table 5 brainsci-07-00080-t005:** Average YGTSS and UPDRS scores of the sample.

	Average Scores	Standard Deviation
YGTSS	46.14	14.07
UPDRS	89.81	26.38

**Table 6 brainsci-07-00080-t006:** Average scores of the sample in the four tasks and average total score.

Diagnosis	Average Score Task 1 ± S.D.	Average Score Task 2 ± S.D.	Average Score Task 3 ± S.D.	Average Score Task 4 ± S.D.	Average Total Score ± S.D.
Parkinson Disease	93.6 ± 7.4	95.5 ± 8.2	97.9 ± 9.8	97.0 ± 10.9	94.4 ± 10.0
Tourette Syndrome	99.7 ± 9.7	102.1 ± 8.3	103.9 ± 13.1	101.4 ± 9.2	103.1 ± 13.0
Sig. Asint. Two-sided	0.011 *	0.003 *	0.094	0.129	0.019 *

* = the scores of the two groups have a statistically significant difference.

**Table 7 brainsci-07-00080-t007:** Results of the sample in score ranges.

Contingency Table					
			Diagnosis		Total
			Parkinson Disease	Tourette Syndrome	
Score range	Score range 1 below 93	Counting	14	5	19
		%within totscore3	73.7%	26.3%	100.0%
		%within diagnosis	51.9%	18.5%	35.2%
	Score range 2 between 93 and 107	Counting	10	14	24
		%within totscore3	41.7%	58.3%	100.0%
		%within diagnosis	37.0%	51.9%	44.4%
	Score range 3 over >107	Counting	3	8	11
		%within totscore3	27.3%	72.7%	100.0%
		%within diagnosis	11.1%	29.6%	20.4%
Total		Counting	27	27	54
		%within totscore3	50.0%	50.0%	100.0%
		%within diagnosis	100.0%	100.0%	100.0%

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
