# Peer review of "Creativity Assessment in Subjects with Tourette Syndrome vs. Patients with Parkinson’s Disease: A Preliminary Study"

_brainsci, 2017, doi:10.3390/brainsci7070080_

Round 1

Reviewer 1 Report

Thank you for conducting this important research. There is a considerable need for research in this area, and a number of different practitioners and disciplines will be interested in this study.  It will be important to revise the manuscript and to get this work into publication promptly. That said, there are a number of areas for improvement in the articulation of the methods, the description of limitations and the implications of the work, as well as the general editing/presentation. 

Firstly, it will be worthwhile to do a bit more of an update on the most recent literature in both the clinical areas (TS and PD) as well as the notion of creativity assessment and treatment.  

Further, the description of the measurement (the ASK) requires much more description and justification for the way the ASK was modified.  For example it will be important to note if others have modified the standardized test in quite this way, and to hear more of the potential limitations of using the measure in this manner.  It seems the choice to adapt this measure has very practical rationale, however this must be explored in greater detail for the strengths/limits etc. 

While some of the limitations are explained, the link to an appropriate sample size is likely not based on the number of modifications made to the measure. There are a number of challenges to the strength of the results and, the results and implications as written are overstated (that the hypotheses are "confirmed". In addition, the leap to state that "art therapy should improve creativity of TS and TD patients" is not well founded in this study at all, with no exploration of art therapy or effectiveness with these patient groups. 

As such, these key areas of the manuscript (the literature review, description of the methods, the write up of results and discussion of implications/recommendations) will all need revision before publication can be considered. Several grammatical and word choice corrections will also be required for publication of this manuscript.  I look forward to reviewing the next edition of this work.

Author Response

Milan, 14th April 2017

Dear reviewer,

We thank you for their interest in our paper and constructive comments. We have considered all points that you raised. Please find in the cover letter a point-to-point response. In our revised manuscript we highlighted with visible track-changes the made modifications.

We think we have satisfactory replied to all questions and hope for a positive response.

With best wishes,

Carlotta Zanaboni Dina

Mauro Porta

Christian Saleh

Domenico Servello

Reviewer 2 Report

Based upon the assumption that dopamine modulates creativity, the authors chose to test the hypothesis that Tourette patients would score significantly better than PD patients on an objective measure of creativity.  They did find in favor of their hypothesis.  While the premise of the study is novel (and creative), there are several key issues to address before the study could be considered for publication:

1.     The following items should please be corrected in the background section on Tourette syndrome

a.      Line 33: It would be more appropriate to state, “improve dramatically;” remission is a controversial term since 90% of patients who feel they are virtually tic free still have tics (see Papert et al 2003).

b.     Reference 4 is a review paper.  It would be preferable to cite the individual studies (at minimum: Erenberg et al 1987; Leckman et al 1998) or a meta-analysis that covers most longitudinal studies to date.

c.      Statement in Line 40, that TS is characterized by abnormal dopaminergic synaptogenesis is outdated.  The reference 6 cited here is outdated- from 1983!  More recent manuscripts should be cited, and these paint a far more complex picture (where other neurotransmitters including GABA play a primary role, though dopamine may still be involved.)  The F1000 review by Jankovic might be a good place to start.

d.     Line 50- It is stated that tetrabenazine is an inhibitor of dopamine reuptake.  This drug blocks presynaptic vesicular storage (reduces dopamine availability) and is NOT a reuptake inhibitor.  Please see F1000 review by Jankovic (or earlier open label papers on use of this drug for hyperkinetic disorders) for review of its mechanism of action.

2.     The following items should please be corrected in the background section on Parkinson disease

a.      The authors state that in some cases DBS can precede dopaminergic therapy.  Best to avoid this very controversial statement.  While this may have been done on a case study patient basis, this is far from the norm while treating a patient with Parkinson’s Disease. See http://www.prd-journal.com/article/S1353-8020(15)30047-X/abstract

b.     Please consider discussing evidence that mesocorticolimbic dopamine system in PD is intact relative to striatonigral (as evidenced by high rates of impulse control disorders in patients treated with dopamine agonists, and consider how this supports (or refutes) the theory that a hypodopaminergic state in PD would be expected to influence creativity.

c.      The authors mention a hypothesis that Dopaminergic therapy may encourage creative thinking in some patients with PD, however a recent study the journal of Parkinson’s disease refutes the relationship of dopaminergic treatment to emergence of artistic thinking. Please consider this in the intro or discussion sections- http://content.iospress.com/articles/journal-of-parkinsons-disease/jpd150681

3.     For background section on creativity

a.      Line 91-93 suggest dopaminergic activity in mesocorticolimbic structures may impact creativity. Please consider (as in c above) how dopamine agonists might therefore impact creativity?  Should the current study have considered PD patients taking dopamine agonists separate from those who did not take them?  Along the same line of thinking; should the study have considered Tourette patients taking anti-dopaminergic therapies separately from those who were taking them?

b.     Line 96-98 is grammatically awkward; please address.

4.     For study design

a.      Please consider a and b above. 

b.     Please consider also, should DBS patients have been analyzed separately?

c.      What is the expected variability between normal subjects, and how would this have been used to calculate power and sample size of this trial?

d.     Does creativity differ between age groups in a healthy control population? While a healthy control population is cited (from a separate reference), what were the ages of the sample?  Were they significantly different from the subjects in this trial (if so, please consider studying an age-matched control sample.)

e.      If creativity can change with age, then why were Tourette and PD subjects not matched for age?  The mean age between the groups was significantly different.

f.       How did the DBS targets (and settings) for PD and Tourette subjects differ?  Why were DBS patients included, if targets and settings differ so much?

5.     For discussion section

a.      Please consider framing this study as a pilot study (and propose further research that is indicated by the findings)

b.     Please expand the limitation section to include all limitations of the sample once the study is revised further. These may included, but are not limited to

                                                              i.      Failure to age-match

                                                           ii.      Heterogeneous sample in terms of number of subjects with DBS (and lead location/DBS settings), age, types of medication

                                                         iii.      Lack of comparison with an age-matched sample (ideally the study will not be resubmitted for publication if this limitation still exists)

Author Response

(The authors gave the same response as above.)

Round 2

Reviewer 2 Report

This reviewer’s primary concerns have been addressed.

In order to ensure that conclusions of the study are clearly and correctly stated

the authors should please consider (or editors should require or arrange) external review of grammatical content before finalizing the manuscript for publications

Some examples where choice of words is still grammatically awkward include:

Line 66 “refuse drug assumption” is an incorrect use of the word assumption

Line 67 “they usually opt from a creative job” (should be “usually opt for”)

Line 70 “Parkinson disease is a holistic pathology and” (“holistic” doesn’t fit here.)

Line 71 “Patients with PD are 4.1-4.6 million worldwide 
” (perhaps state, “PD impacts an estimated 4.5 million people across the world’s 10 most population countries”)

Line 131 “creativity is a common matrix” (unclear to this reader what that means)

Line 167 abbreviation “Tab” is incorrectly used for “table”

Line 300 abbreviation “ST” is incorrectly used in place of “TS”

Author Response

Milan, 30th April 2017

Dear Reviewer,

We thank you for your interest in our paper and constructive comments. We have considered all points that you raised in the minor revision. Please find in the Cover Letter a point-to-point response. In our revised manuscript we highlighted with visible (Christian Saleh’s comments in underlined light blue) track-changes the made modifications.

We think we have satisfactory replied to all questions and hope for a positive response.

With best wishes,

Carlotta Zanaboni Dina

Mauro Porta

Christian Saleh

Domenico Servello